# A New MiRNA MiRm0002 in Eggplant Participates in the Regulation of Defense Responses to Verticillium Wilt

**DOI:** 10.3390/plants10112274

**Published:** 2021-10-23

**Authors:** Wenjiao Zhu, Xinru Liu, Min Chen, Nianjiao Tao, Alexander Tendu, Qing Yang

**Affiliations:** College of Life Sciences, Nanjing Agricultural University, Nanjing 210095, China; liuxinru2019@126.com (X.L.); chenmincm@njau.edu.cn (M.C.); nianjiaotao@163.com (N.T.); aatendu@yahoo.com (A.T.)

**Keywords:** *Solanum melongena* L., miRm0002, Verticillium wilt, defense response

## Abstract

Verticillium wilt is a major disease that severely affects eggplant production, and a new eggplant miRNA named miRm0002 identified through high-throughput sequencing was highly induced by Verticillium wilt infection. However, the miRm0002 function was still elusive. In this study, the sequence of the miRm0002 precursor was cloned and transgenic eggplants were constructed. In vivo inoculation test and in vitro fungistatic test showed that overexpressing miRm0002 lines were more resistant to *Verticillium dahliae* and inhibiting miRm0002 lines were more sensitive, compared to the wild-type (WT) control. Some physiological indicators were selected and the results showed that SOD, POD, and CAT activities were significantly increased in Verticillium wilt-infected overexpressing miRm0002 lines, indicating that the expression of miRm0002 activates the antioxidant system. QRT-PCR assay showed that the transcript expression of miRm0002 candidate target *ARF8*, a gene encoding auxin response factor was negatively related to miRm0002 in WT as well as transgenic eggplants. However, RLM-RACE mapping and degradome sequencing showed miRm0002 could not cleave the sequence of *ARF8*. Taken together, these data suggest that miRm0002 plays a positive role in the defense response of eggplant against Verticillium wilt.

## 1. Introduction

Verticillium wilt (V. wilt), a disease caused by soil-borne *Verticillium dahliae*, reduces up to 34.1–42.5% of eggplant production worldwide [1,2]. Over the last two decades, significant progress has been made by scientists and farmers in improving the resistance of eggplants to V. wilt. Introducing resistance genes such as *StoVe1*, *StoL13a*, *StoNPR1*, and *StoCYP77A2* from the highly resistant wild eggplant *Solanum torvum* into potato or tobacco has significantly improved their resistance to V. wilt [3,4,5,6]. The transcriptome analysis of the wild eggplant species *Solanum*
*aculeatissimum* resistant to V. wilt identified 11,696 upregulated and 5949 downregulated genes, which are candidates for further genetic improvement [7]. All these studies only focused on screening or introducing the resistance genes, however, the molecular mechanism of how eggplant interacts with the disease agent fungus remains elusive.

MiRNAs are a series of non-coding regulatory RNAs that play important roles in plant growth and development, and inhibit the expression of target genes by cleavage of target transcripts or inhibition of translation. Previous studies have showed that some miRNAs are involved in regulating the defense responses of plants against pathogen attack. For instance, in *Arabidopsis thaliana*, the expression of miR393 whose target is an auxin response factor, was induced by a short peptide flg22, restricting the growth of bacteria [8]. However, *Arabidopsis* miR398 and miR773, which are down-regulated by flg22, negatively regulate PTI resistance to *P. syringae* [9]. In tobacco, nta-miR6019 and nta-miR6020 can inhibit the expression of the Toll and Interleukin-1 receptor-NB-LRR immune receptor N that confers resistance to tobacco mosaic virus [10]. In tomato, the expression of NBS-LRR defense genes upon infection by *P. syringae* is induced by suppressing the miR482-mediated silencing cascade [11]. In cotton, suppressing the miR482-mediated gene silencing pathway induces expression of NBS-LRR defense genes upon attack of *V. dahliae* [12]. In eggplant, small RNA deep sequencing identified many miRNAs in responsive to *V. dahliae* infection, like miR156 [13]. Overexpressing miR395 increased eggplant sensitivity to *V. dahliae* infection [14]. In potato, over-expression of miR482e enhanced plant sensitivity to *V. dahliae* infection [15]. 

The phytohormone auxin has also been identified as a key regulator in plant-pathogen interactions, and plays a myriad of roles in plant resistance to pathogens: auxin biosynthesis and accumulation cause host plant cell wall expansion and stomatal opening, which promote disease susceptibility. Camalexin biosynthesis and JA/ethylene signaling induced by auxin signaling promote resistance to necrotrophic pathogen; suppressed indole glucosinolates (IGs) biosynthesis and SA signaling by auxin signaling reduce the resistance to biotrophic pathogens [16,17]. Moreover, overexpression of AFB1, an auxin signal receptor, renders the plant more susceptible to biotrophic pathogens due to the reduction in IG accumulation and SA levels mediated by ARF1 and ARF9 [18]. Many elements involved in auxin synthesis or signaling pathway are regulated by miRNAs: auxin signal receptors TIR1 and AFB1/2/3 are targets of miR393 [8], and different auxin response factors are regulated by different miRNAs, for example, *ARF6* and *ARF8* are targets of miR167, and *ARF10*, *ARF16*, and *ARF17* are targets of miR160a [19,20].

Previous work prepared two sets of eggplant small RNA libraries from mock-infected and *V. dahlia*-infected seedlings of eggplants, and identified two nascent miRNAs namely miRm0001 and miRm0002 through high-throughput sequencing, with miRm0002 being significantly induced following pathogen infection [13]. However, the role of miRm0002 in regulating eggplant response to *V. dahliae* is still elusive. In this study, the disease resistance of miRm0002 overexpression and inhibitory expression transgenic lines to *V. dahliae* was systematically examined and the expression of potential target *ARF8* (an auxin response factor) was examined, demonstrating that miRm0002 enhances eggplant resistance to V. wilt, possibly by translational repression not cleavage of *ARF8*.

## 2. Results

### 2.1. The Sequence, Stem Loop Structure, and Gene Clone of the MiRm0002 Precursor

According to the predicted sequence of miRm0002 precursor (Figure 1a) [13], the stem loop structure was folded by applying the online mfold software (http://www.unafold.org/mfold/applications/rna-folding-form.php accessed on 20 September 2021), and the results showed that the 19 nt of miRm0002 was located at the stem (Figure 1b). 194 bp pri-miRm0002 sequence was then cloned for further research (Figure 1c).

### 2.2. Production of Transgenic Eggplant

To investigate whether miRm0002 is involved in the defense response of eggplant to V. wilt, the sense and antisense sequences of miRm0002 precursor were cloned and introduced into pCAMBIA1304 expression vectors driven by the CaMV35S promoter (Figure 2a), which were finally transferred into eggplant cultivar Suqi (Figure 2b). Molecular identification performed by PCR with primer pair miRm0002_F and GFP_R (Appendix A) and sequencing showed that the expected 247 bp product contains the partial sequence of both precursors and pCAMBIA1304 vector. mO lines were obtained namely mO1, mO2, mO3, and mO4 (Figure 2c) as well as 4 mI lines namely mI1, mI2, mI3, and mI4 (Figure 2d). Stem loop qRT-PCR of miRm0002 revealed that the expression of mature miRm0002 was increased in the mO transformants while it was decreased in the mI transformants (Figure 2e).

### 2.3. Disease Resistance of Transgenic Eggplant to V. Wilt

To understand the resistance of transgenic eggplant to V. wilt, the in vitro anti-fungal assay was performed with the crude protein extract from the transgenic lines obtained above (Figure 3). Only mO4 and mI4 transgenic lines are presented (Figure 3a) because there was no significant difference between transgenic lines with the same transgenic constructs by analyzing the inhibition rates (Figure 3b). The results showed that the antifungal efficacy of mO lines was twice higher than that of control. However, the antifungal efficacy of mI lines was not readily apparent, showing slight inhibition compared with that of PBS (Figure 3). The wild-type control and eggplant transformed with vector pCAMBIA1304 had no difference between each other but had a basal antifungal efficacy which differed from the PBS sample.

To further assess the in vivo resistance of the transgenic lines, the infection experiment of *V. dahliae* was conducted with mO, mI lines and control lines at eggplant seedling stage (Figure 4). The phenotype of wilting and the statistical analysis of disease index showed that mO lines were more resistant to V. wilt while the mI lines were more susceptible compared to the control lines (Figure 4a,b). Further, in order to know whether expression of miRm0002 affects the propagation of pathogens in transgenic eggplants, quantitative detection of fungal biomass was carried out by semi-quantitative PCR of *V. dahliae* internal transcribed spacer (ITS) in the vascular tissue of the eggplants. The results showed that after infection for three days, the ITS DNA level was lower in the mO plants and higher in the mI plants, compared to the control plants (Figure 4c). This suggests that overexpression of miRm0002 inhibits the proliferation of *V. dahliae* in eggplant.

### 2.4. Antioxidant Activity in Transgenic Eggplants

When plants are infected by pathogens, they experience an accumulation of oxygen free radicals, which leads to the activation of the antioxidant system hence increasing the activity of antioxidant enzymes. In this study, the activity of SOD, POD, and CAT was systematically examined in mO, mI, and control lines, respectively (Figure 5). The results showed that infection of *V. dahliae* increased the accumulation of SOD, POD, and CAT in all overexpressing lines and after *V. dahliae* treatment, overexpression of miRm0002 significantly increased SOD, POD, and CAT compared to both controls (Figure 5a–c). However, there is no obvious difference in activities of SOD, POD, and CAT in all inhibiting lines, whether before and after treatment or compared with the control (Figure 5a–c).

### 2.5. Analysis on Predicted Targets of MiRm0002 

MiRNAs exert their regulatory role by down-regulating the expression of their target genes at the post-transcriptional level. The target gene for miRm0002 was predicted using the online software psRNATarget (http://plantgrn.noble.org/psRNATarget/ 20 September 2021) and two candidate sequences in eggplant were obtained: TC5469 and TC9181, of which only TC5469 was a complete known sequence encoding the auxin response factor ARF8 involved in auxin signal transduction [21]. Therefore, the expression of miRm0002 and *ARF8* was quantified in control, mO, and mI eggplants under both water and *V. dahliae* treatments, and the results showed that the expression regulation trends of miRm0002 and *ARF8* are opposite in different plants, which indicated that *ARF8* expression was downregulated by miRm0002 cleavage or *V. dahliae* treatment (Figure 6a). To further confirm the expression regulation mode of *ARF8*, RLM-RACE assay was performed showing that the cleavage site of target mRNA *ARF8* was 50 bp upstream of matching sites in large proportion (Figure 6b). Besides, the degradome sequencing showed that there was no corresponding *ARF8* mRNA fragment [22]. All these results showed that miRm0002 could not cleave the sequence of *ARF8* at the matched site.

## 3. Discussion

MiRNAs play their specific roles in regulating the defense responses against pathogen attack in different plants. In the case of *V. dahliae*, suppression of cotton miR482 promoted resistance upon attack [12] while overexpressing potato miR482 enhanced susceptibility to *V. dahliae* infection [15]. In eggplant, small RNA deep sequencing identified many miRNAs in response to *V. dahliae* infection, like miR156 and miR482, which are induced by infection, and others like miR393, miR395, and miR399 whose expressions are gradually inhibited with increased infection time [13]. Furthermore, two new miRNAs namely miRm0001 and miRm0002 were identified, miRm0002 being induced by *V. dahliae* infection, which indicated that it might be involved in defense against V. wilt. To test this hypothesis, this study comprehensively analyzed the characteristics and functions of the miRNA though genetic engineering, physiological and molecular methods, and all the results showed that miRm0002 is involved in eggplant defense response and its overexpression can enhance the resistance of eggplant to V. wilt, which indicated that *V. dahliae*-induced miRm0002 expression enhanced the eggplant resistance.

MiRNAs exert their function by directly targeting mRNA of their target genes for post-transcriptional repression. Therefore, the identification of the target is an important part of miRm0002 research. So far, there are four accepted methods for miRNA target gene validation: verification of the cleavage site by 5′-RACE [23], mutation of the site of miRNA to analyze the relationship between the expression of the two genes [24], the transgenic way [25,26] and degradome sequencing [27]. In these studies, three ways were employed for miRNA target validation. First, the expression analysis of *ARF8* in miRm0002 over-expression and antisense inhibition lines showed the expression of miRm0002 was inversely correlated to *ARF8* in both mO lines and mI lines. However, RLM-RACE assay showed that the cleavage sites are upstream of miRm0002 binding sites, and degradome sequencing used for identification of miRNA targets in eggplant in response to *V. dahliae* [22] could not identify the fragment of *ARF8* at the matched site, which indicated that miRm0002 may regulate the expression of *ARF8* by translation inhibition not by cleavage. However, that needs to be further investigated. 

It is worth noting that in previous target gene prediction, both *ARF8* and *NTF3* (mitogen-activated protein kinase homolog gene) were the candidate target gene of miRm0002. However, *NTF3* could not be analyzed due to the lack of a complete sequence. For the same reasons there may be still some target genes which have not been found. Therefore, this suggests that miRm0002 may mediate also different regulation and participate in different biological processes.

## 4. Materials and Methods

### 4.1. Plants Material and Growth Conditions

The eggplant cultivar Suqi (*Solanum melongena* L) was obtained from Jiangsu Province Academy of Agricultural Sciences. Seeds were sterilized and inoculated on solid MS medium [28] containing 3% sucrose and 0.8% agar at pH 5.8 with 200 mg/L cefotaxime (Amresco, USA) for further sterilization, and maintained under 28 ± 2 °C dark condition for 5 d for germination. They were then cultured in Petri-dishes containing solid MS medium under a 28 °C 16 h light/20 °C 8 h dark regime for 4 weeks. Subculture was done once every 4 weeks. In vitro plantlets were used for transformation experiment, or they were sown in pots with a potting mix of 1:1:1 ratio of loam, vermiculite, and perlite and grown in greenhouse for gene cloning, expression analysis, and pathogen infection experiment [13]. 

### 4.2. Pathogen Culture and Infection

A highly aggressive VDT 18 strain of *V. dahlia* [5] isolated from eggplant with V. wilt was kindly provided by The Plant Protection College of Nanjing Agricultural University (China) and was cultured on potato dextrose agar (PDA) plate in darkness at 25 °C for 14 days to collect spores, which were washed with distilled water to a concentration of 5 × 10^7^ spores mL^−1^ or in Czapek’s culture medium (NaNO_3_ 2%, K_2_HPO_4_ 1%, MgSO_4_·7H_2_O 0.5%, KCl 0.5%, FeSO_4_·7H_2_O 0.01%, and sucrose 3%) to obtain crude toxin, which was filtered with 8 layers gauze and the concentration was adjusted to 8 mg mL^−1^.

The inoculation of Verticillium wilt was carried out by root irrigation. Eggplant seedlings with 4~5 true leaves grown were chosen for infection with the mixture of spores and crude toxin (1:1) of *V. dahliae* which was poured into their culture medium of each pot until saturated [29]. Mock-infected plants were inoculated with sterile distilled water. After infection for 3 days, equal amounts of leaves were harvested and immediately frozen in liquid nitrogen for RNA extraction. 

### 4.3. RNA Extraction and Expression Analysis

Total RNA was extracted from the leaves of eggplant plants using Takara RNA Extraction kit. The first-strand cDNA was then synthesized with specific Stem-loop RT primer miRm0002_RT and U6_R for miRNA expression (Appendix A) [30], or with Oligo(dT)18/Random Primer for expression of target gene *ARF8* using Super RT kit (Bio Teke, Wuxi, China), and then used as template for qRT-PCR analysis. Primer sequences for *miRm0002*, *ARF8,* and their internal references are listed in Appendix A. QRT-PCR was performed in triplicate using SYBR PremixExTaq reaction system (TaKaRa Biotech, Japan) on the ABI PRISM7500HT FAST Real-Time PCR System (Applied Biosystems, Foster City, CA, USA) and *U_6_* was used as a control to normalize the level of total RNA for miRm0002 measurements, and *EF-1α* was used as internal reference for *ARF8* analysis. 

### 4.4. RACE Mapping of miRNA Target Cleavage Sites

Total RNA was extracted from leaves of eggplant Suqi seedlings in four-leaf stage using the Trizol reagent (Sangon, Shanghai, China), then it was directly ligated to the RNA 5′ adaptor (5′-GCTGTCAACGATACGCTACGTAACGGCATGACAGTGCCCCCCCCCCCCCCC-3′) and 3′ and reverse-transcribed using M-MuLV First Strand cDNA Synthesis Kit (B532435) with oligo (dT) primer. The cDNA samples were amplified by nested PCR according to invitrogen 5′ race system manual. Initial PCR was carried out using the 5′ outer primer (5′-GCTGTCAACGATACGCTACGTAACGGCATGACAGTGCCCCCCCCCCCCCCC-3′) and gene-specific outer primer (5′-TTCTAGCCCAAACATGCGAGCAAGC-3′). Nested PCR was carried out using 1 μL of the initial PCR reaction, and 5′ nested primer (5′-GCTGTCAACGATACGCTACGTAAC-3′) and gene-specific inner primer (5′-GAGTGACCGTCCAAAGGACCCTGATT-3′). After amplification, 5′-RACE products were gel purified and cloned into the pMD180T vector and the clones were sequenced.

### 4.5. Vector Construction and Plant Transformation

A cDNA library from eggplant (Suqi) was used for amplifying the Pre-miRm0002 sequences using gene-specific primers (Appendix A), and the amplified fragments were cloned into the vector pCAMBIA1304 to construct the miRm0002 over-expression vector pCAMBIA1304-pre-miRm0002 and antisense inhibition vector pCAMBIA1304-antisense pre-miRm0002 respectively. The resulting constructs were introduced into *Agrobacterium tumefaciens* strain GV3101 using the freeze-thaw method [31].

The stem segment explants of in vitro plantlets were pre-cultured for 2 days, then infected with GV3101 containing plasmid pCAMBIA1304-pre-miRm0002 and pCAMBIA1304-antisense pre-miRm0002 respectively. After co-culturing for 2 days, the explants were transferred to the differentiation medium for callus formation and shoot regeneration. Transgenic plants were selected on media containing 10 mg/L^−1^ Hygromycin B (Amresco, USA) by root screening [3].

### 4.6. Transgenic Plant Identification

Genomic DNA was extracted from tender leaves of the transgenic and control plants (transformed with vector only) using the CTAB method. Molecular identification was carried out by PCR using gene-specific forward primer and vector-specific reverse primer (GFP_R, Appendix A).

### 4.7. Evaluation of Plant Disease Resistance

Plant disease resistance was evaluated with disease phenotype and disease index. Disease phenotype was investigated on the third day post-inoculation and disease index was calculated according to the disease grading criterion of grades 0–4 defined by Liu et al. (2012) [3] using the formula: disease index = [∑ (number of diseased plants × disease grade)/(total number of investigated plants × the highest disease grade)] × 100. 

### 4.8. Quantitative Analysis of V. dahliae in Tissues

Quantitative analysis of *V. dahliae* in plant tissue was done with reference to Pantelides et al. [32]. The above-ground parts were cut at the soil level. All samples were taken from the middle part of the above-ground portion of the plantlets, then rinsed with distilled water, and ground into powder in liquid nitrogen. Total DNA was extracted using CTAB method for semi-qPCR assay. The quantification of *V. dahliae* was conducted by measuring the DNA levels of *V. dahliae* internal transcribed spacer (ITS), which used eggplant *α-**t**ub**u**lin* as internal standards to normalize the DNA template amounts. Primer sequences are listed in Appendix A.

### 4.9. In Vitro Anti-Fungal Assay

In vitro anti-fungal efficacy of crude protein extracts from transgenic and control eggplants was carried out according to the improved mycelium growth inhibition method [33]. The anti-fungal activity of the sample was indicated by the radius of the inhibition zone. Circular wells were drilled at the center of the solid PDA in an 80-mm Petri-dish and two other positions equidistant from the center on opposite sides. For all Petri dishes, *V. dahliae* strain blocks with the same vigorous growth were inoculated in the central well, and 32 µg crude protein extract was added to the other two wells. Phosphate buffered saline (PBS) was used as negative control. The antifungal efficiency was calculated as follows: antifungal efficiency % = [(Colony diameter of *V. dahliae* in PBS control-Colony diameter of *V. dahliae* in test extract)/(Colony diameter of *V. dahliae* in PBS control)] × 100%. 

### 4.10. Antioxidant Enzyme Assay

Enzyme solution for antioxidant assay was extracted using the method of Wan et al. [34], and protein concentration was determined according to the method of Bradford [35] using calf serum (BSA) as the standard. The activity of superoxide dismutase (SOD) was determined by measuring its ability to inhibit the photochemical reduction of NBT [36]. One unit of SOD activity (U) was defined as the amount of crude enzyme extract required to inhibit the maximum reduction rate of NBT by 50%. The activity of peroxidase (POD) was determined by measuring the oxidation of guaiacol (extinction coefficient 26.6 mM^−1^ cm^−1^) at 470 nm [37]. Catalase (CAT) can catalyze the decomposition of H_2_O_2_ into oxygen and water, and the enzyme activity was determined by the H_2_O_2_ decomposition rate at 240 nm [38].

### 4.11. Statistical Analysis

The data were analyzed by one-way ANOVA using IBM SPSS Version 19.0. Displayed results are means ± SD (error bar) for two biological replicates and three technical replicates. Asterisks indicate the significance level in one-way ANOVA (*, *p* < 0.05; **, *p* < 0.01).

## 5. Conclusions

In summary, this study reveals the positive regulation of miRm0002 in the defense response of eggplant V. wilt by in vivo inoculation test, in vitro fungistatic test, measurement of some physiological indicators in mO, mI lines, and controls. All these results provide not only important insights into the function and mechanism of miRm0002 in eggplant defense against V. wilt, but also theoretical basis for eggplant V. wilt resistance breeding. 

## Figures and Tables

**Figure 1 plants-10-02274-f001:**
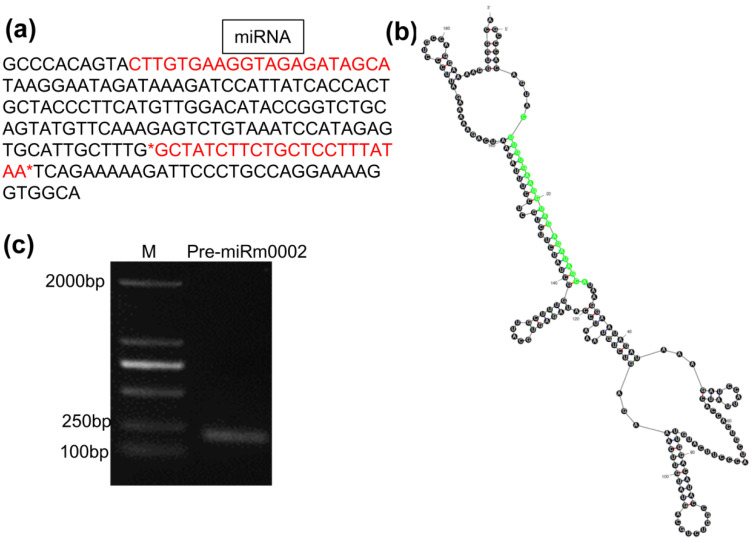
The sequence (**a**), stem loop structure (**b**), and gene clone (**c**) of the miRm0002 precursor. M: DL2000; pre-miRm0002: the precursor of miRm0002.

**Figure 2 plants-10-02274-f002:**
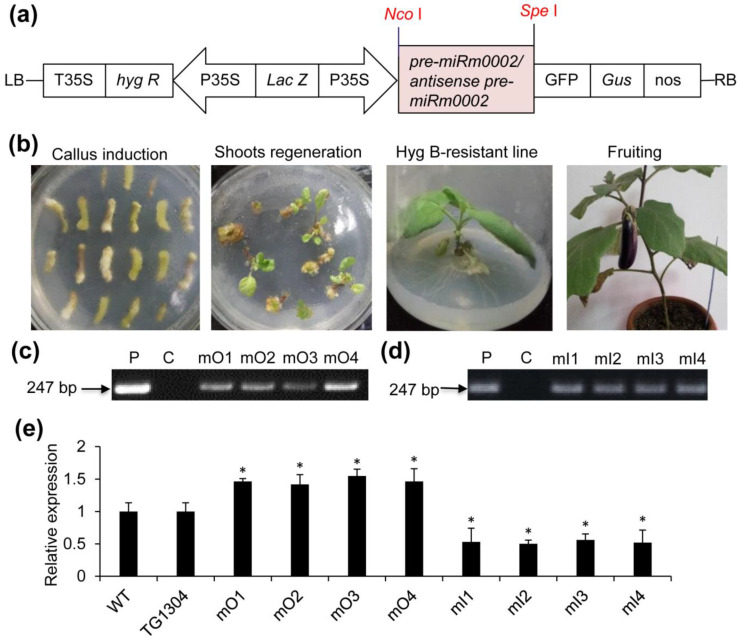
Construction of 35S: pre-miRm0002 and antisense pre-miRm0002 vectors respectively and identification of transgenic eggplants by PCR. (**a**) Schematic representation of two vectors pCAMBIA1304-pre-miRm0002 or antisense pre-miRm0002 with sites of restriction enzymes, respectively; (**b**) Regeneration of transgenic eggplants. PCR-based identification of miRm0002 overexpression lines (**c**) and miRm0002 inhibition expression lines (**d**). P: pCAMBIA1304-pre-miRm0002 vector or pCAMBIA1304-antisense pre-miRm0002 vectors, C: non-transformed plants, mO: miRm0002 overexpression transgenic lines 1–4, mI: miRm0002 inhibition expression transgenic lines 1–4. (**e**) QRT-PCR-based analyses of miRm0002 transcripts. WT: non-transformed plants, TG1304: Control plants (transformed with vector only). The expression level of miRm0002 in non-transformed plants was considered as the background level and set to 1. *, *p* < 0.05.

**Figure 3 plants-10-02274-f003:**
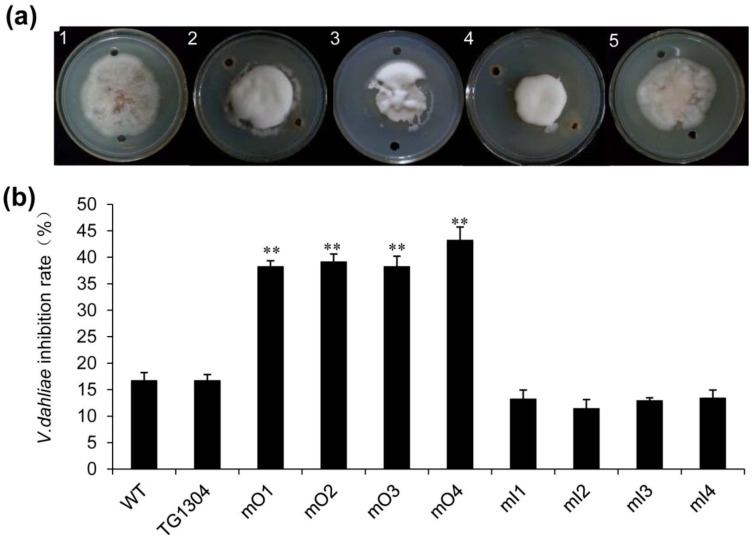
Analysis of the anti-fungal efficacy of transgenic eggplant protein extracts. (**a**) Growth of *V. dahliae* on the medium containing the extract of total proteins from eggplant. 1: PBS control, 2: wild type eggplant, 3: control plants (transformed with 1304 vector only), 4: only mO4 transgenic line, 5: only mI4 transgenic line. (**b**) *V. dahliae* inhibition rate of different transgenic lines. **, *p* < 0.01.

**Figure 4 plants-10-02274-f004:**
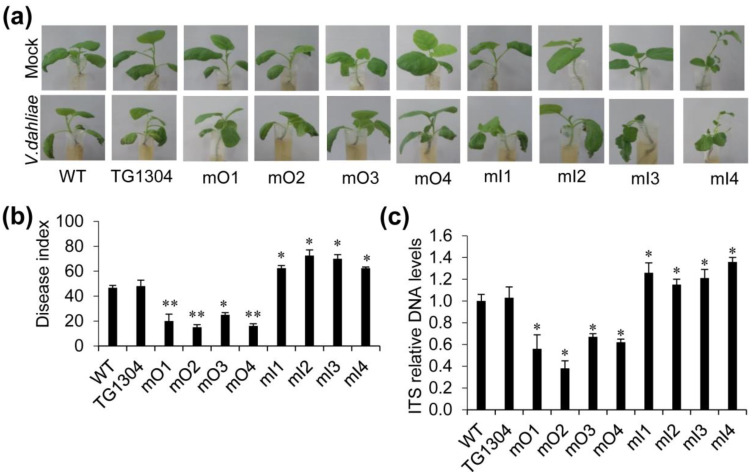
Analysis of the resistance of transgenic eggplants against V. wilt. (**a**) Two-week-old transformants and the control plants (both WT and TG1304) were inoculated with *V. dahliae* (mock treatments were done with water) and photographed at 3 days. (**b**) The disease index was analyzed. (**c**) PCR of fungal colonization by comparing *V. dahliae* ITS DNA levels relative to eggplant *α-**tub**ulin* DNA level (for equilibration). Expression level of ITS in the control plants was considered as background level and set to 1. *, *p* < 0.05; **, *p* < 0.01.

**Figure 5 plants-10-02274-f005:**
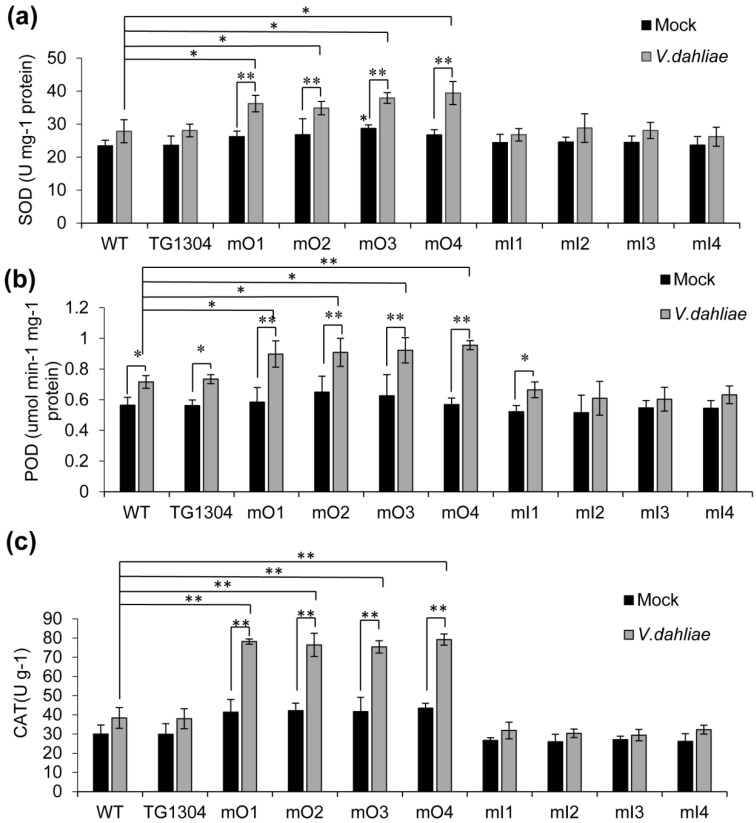
Antioxidant activity analysis of transgenic eggplants. Total activity of SOD (**a**), POD (**b**), and CAT (**c**) in the leaves of two-week-old mO and mI transgenic lines and the control plants (both WT and TG1304) upon mock inoculation or *V. dahliae* infection for 3 days. *, *p* < 0.05; **, *p* < 0.01.

**Figure 6 plants-10-02274-f006:**
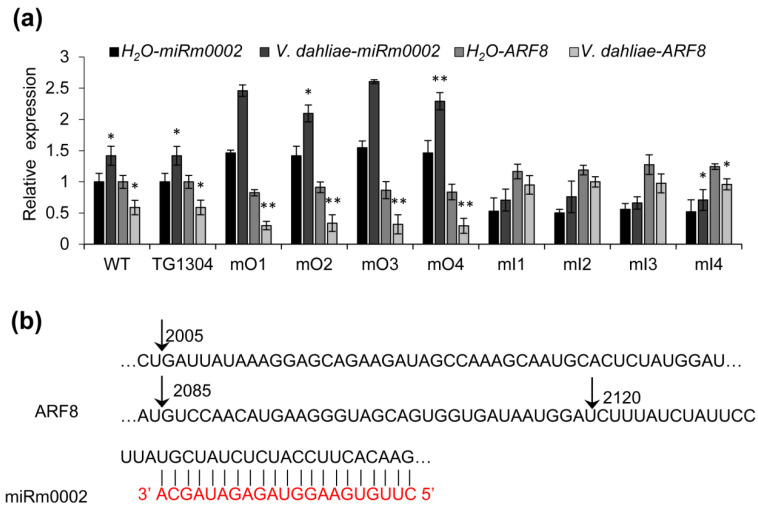
Analysis on predicted targets of miRm0002. (**a**) qRT-PCR analysis of miRm0002 and *ARF8* of mO and mI transgenic lines and the control plants (both WT and TG1304) upon mock inoculation or *V. dahliae* infection. *, *p* < 0.05; **, *p* < 0.01. (**b**) The cleavage sites in the *ARF8* mRNA determined by RLM-RACE, among which the frequency of 5’-RACE clones corresponding to the 2085 bp site is high.

## Data Availability

The original data sets described in the study are included in the article/Appendix A. Further inquiries can be addressed to the corresponding author.

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
