# Peer review of "A New MiRNA MiRm0002 in Eggplant Participates in the Regulation of Defense Responses to Verticillium Wilt"

_plants, 2021, doi:10.3390/plants10112274_

Round 1
Reviewer 1 Report
A manuscript titled “A new miRNA miRm0002 in eggplant participates in the regulation of defense responses to Verticillium wilt” is devoted to the role of eggplant miRNA named miRm0002 in interaction with Verticillium wilt infection. The miRm0002 precursor was cloned and transgenic eggplants overexpressing miRm0002 (mO) and inhibiting miRm0002 (mI) were obtained. In vivo inoculation test and in vitro bacteriostatic test showed that mO lines were more resistant to Verticillium dahliae and mI lines were more sensitive, compared to the wild-type (WT) control. SOD, POD and CAT activities were significantly increased in V. wilt-infected mO lines, indicating that the expression of miRm0002 activates the antioxidant system. The transcript expression of miRm0002 candidate target ARF8, a gene encoding auxin response factor was negatively related to miRm0002 in WT as well as transgenic eggplants. Thus, all the data suggest that miRm0002 plays a positive role in the defense response of eggplant against V. wilt.
This is a well-written and very enthusiastic work. It can be published in present form.
Author Response
Thanks for the reviewer's comments and suggestions.
Reviewer 2 Report
This is a very good paper that presents a miRNA that has significant impact on V. dahliae, Verticillium wilt in eggplant, and response of relevant disease response pathways. The dynamics associated with miRm0002 are opposite from what has been found with some other miRNA, which makes this work particularly significant. Also the impact of miRm0002 on mRNA was not due to cleavage, which is also unusual, and adds to our overall body of knowledge of mechanism for miRNA (though the exact mechanism is not totally understood in this case). I have added some minor edits in the attached file. I would like a more thorough description of the inoculum preparation and plant inoculation procedure.

Author Response
Thanks for your comments. For minor edits mentioned in the attached file, we revised all minors in the revised manuscript except the following: You have only one citation listed, so the sentence should start with "This study" (page 1). For this, we cited [3-6] for introducing the resistance genes and [7] for screening the resistance genes, therefore, we described “All these studies” nor “This study” here.
Besides, for “more thorough description of the inoculum preparation and plant inoculation procedure” (Was the conidia + crude toxin added to the surface of the pots and watered in, or were holes made to the potting mixture and then a certain number of mls added near the roots? Was the initial mixture of spres poured through cheesecloth to remove mycelia, or was a combination of mycelia and spores added to the pots?), we added more information in the methods of the revised manuscript. The spores were washed with distilled water to a concentration of 5 x 107 spores ml-1, and crude toxin was filtered with 8 layers gauze and the concentration was adjusted to 8 mg ml-1. Then, the mixture was poured into the culture medium until saturated by root irrigation.